# Solar Hydrogen Production and Storage in Solid Form: Prospects for Materials and Methods

**DOI:** 10.3390/nano14191560

**Published:** 2024-09-27

**Authors:** Kathalingam Adaikalam, Dhanasekaran Vikraman, K. Karuppasamy, Hyun-Seok Kim

**Affiliations:** 1Millimeter-Wave Innovation Technology Research Center, Dongguk University-Seoul, Seoul 04620, Republic of Korea; kathu@dongguk.edu; 2Division of Electronics and Electrical Engineering, Dongguk University-Seoul, Seoul 04620, Republic of Korea; v.j.dhanasekaran@gmail.com (D.V.); karuppasamyiitb@gmail.com (K.K.)

**Keywords:** hydrogen energy, solar hydrogen, water-splitting, hydrogen storage, solid hydrogen storage, photoelectrochemical water-splitting, hydrogen fuel cells

## Abstract

Climatic changes are reaching alarming levels globally, seriously impacting the environment. To address this environmental crisis and achieve carbon neutrality, transitioning to hydrogen energy is crucial. Hydrogen is a clean energy source that produces no carbon emissions, making it essential in the technological era for meeting energy needs while reducing environmental pollution. Abundant in nature as water and hydrocarbons, hydrogen must be converted into a usable form for practical applications. Various techniques are employed to generate hydrogen from water, with solar hydrogen production—using solar light to split water—standing out as a cost-effective and environmentally friendly approach. However, the widespread adoption of hydrogen energy is challenged by transportation and storage issues, as it requires compressed and liquefied gas storage tanks. Solid hydrogen storage offers a promising solution, providing an effective and low-cost method for storing and releasing hydrogen. Solar hydrogen generation by water splitting is more efficient than other methods, as it uses self-generated power. Similarly, solid storage of hydrogen is also attractive in many ways, including efficiency and cost-effectiveness. This can be achieved through chemical adsorption in materials such as hydrides and other forms. These methods seem to be costly initially, but once the materials and methods are established, they will become more attractive considering rising fuel prices, depletion of fossil fuel resources, and advancements in science and technology. Solid oxide fuel cells (SOFCs) are highly efficient for converting hydrogen into electrical energy, producing clean electricity with no emissions. If proper materials and methods are established for solar hydrogen generation and solid hydrogen storage under ambient conditions, solar light used for hydrogen generation and utilization via solid oxide fuel cells (SOFCs) will be an efficient, safe, and cost-effective technique. With the ongoing development in materials for solar hydrogen generation and solid storage techniques, this method is expected to soon become more feasible and cost-effective. This review comprehensively consolidates research on solar hydrogen generation and solid hydrogen storage, focusing on global standards such as 6.5 wt% gravimetric capacity at temperatures between −40 and 60 °C. It summarizes various materials used for efficient hydrogen generation through water splitting and solid storage, and discusses current challenges in hydrogen generation and storage. This includes material selection, and the structural and chemical modifications needed for optimal performance and potential applications.

## 1. Introduction

Energy is fundamental to human life, the economy, and global development. It underpins all industrial activities and advancements, making it essential for technological progress and overall human well-being. With the exponential growth of the population and its needs, energy consumption has surged dramatically in recent years [1]. This increased consumption, driven by industrial and technological activities, has led to significant environmental pollution due to reliance on fossil fuels. Fossil fuels, while currently a major energy source, are increasingly problematic. They contribute to environmental pollution through greenhouse gas emissions, have finite reserves that will eventually be depleted, and are unevenly distributed across nations [2]. The CO_2_ emissions from fossil fuels are a major concern, as they drive global warming and environmental degradation [3]. The depletion of fossil fuels, coupled with inadequate renewable energy conversion and the growing demand for energy, underscores the urgent need for efficient renewable energy solutions and effective storage methods [4]. To address these challenges and mitigate environmental damage, transitioning to renewable energy sources is crucial for future generations. Among various renewable energy sources, hydrogen energy stands out as a promising alternative for future energy needs [5,6,7]. Over the years, the energy landscape has shifted from coal to oil, then to natural gas, and now toward hydrogen, reflecting advancements in technology and a growing interest in exploring new possibilities.

Hydrogen presents a highly efficient alternative to other renewable energy sources due to its clean and efficient nature [2,8]. Despite its advantages, hydrogen is not widely utilized because it is not readily available in its pure form; instead, it must be extracted from water and hydrocarbons through complex separation processes. Additionally, transporting hydrogen is challenging due to its gaseous state at low temperatures. Although hydrogen is abundant in water and biomass, it must be converted into a usable form. Various technologies are available for hydrogen production, but only a few are environmentally friendly. Recently, solar hydrogen production through photocatalytic (PC) and photoelectrocatalytic (PEC) water-splitting techniques has garnered significant attention. The photocatalytic reaction-based water splitting does not require separate energy sources and does not generate any toxic byproducts. Therefore, this solar hydrogen and solid storage combined technology is an emerging technology for an environmentally friendly approach compared to other methods [9]. These methods offer the potential for low-cost, clean hydrogen production by mimicking the natural photosynthesis process. Solar water splitting, which uses solar energy to produce hydrogen from water, is a renewable and environmentally friendly method. Hydrogen produced via solar water splitting is efficient both economically and energetically. It holds promise as a clean energy source for powering vehicles through hydrogen-based fuel cells. However, efficient hydrogen storage remains a significant challenge. Hydrogen can be stored as a liquid or gas, but each method has drawbacks. Liquefaction is costly, while compressed gas storage involves safety issues due to the high pressures required [10]. Handling gaseous hydrogen is both tedious and hazardous, and liquid hydrogen requires cryogenic tanks maintained at temperatures below 50 K. Common storage systems use high-pressure gas cylinders with pressures ~20 MPa, but these present safety concerns due to hydrogen’s high flammability. To address these issues, solid-state hydrogen storage offers a promising solution. Storing hydrogen in solid form can mitigate the risks associated with handling liquid or gaseous hydrogen [11,12]. The scientific community is actively exploring solid-state storage media, such as hydrides or porous materials that can absorb hydrogen. These materials can store hydrogen generated from solar energy, addressing future energy needs safely and efficiently. This review consolidates existing research and outlines future developments in hydrogen production and storage. It presents various techniques for hydrogen production and different materials and methods for solid hydrogen storage, highlighting their applications and potential advancements.

## 2. Hydrogen as an Alternative Energy Source

Most of the energy we use today comes from fossil fuels, which are increasingly problematic for global energy concerns [13]. Fossil fuels contribute to environmental pollution through greenhouse gas emissions and are depleted due to over-exploitation driven by population growth and industrial demands. There is an urgent need to replace these harmful conventional sources with efficient alternatives that do not damage the environment. Therefore, finding sustainable, low-cost, and environmentally friendly energy sources is crucial for meeting future energy needs. Several alternative energy sources have been introduced to reduce reliance on fossil fuels and lower greenhouse gas emissions. These include wind, tidal, hydro, and solar energy. Wind energy utilizes wind turbines to generate electricity. While wind energy is renewable and free, it requires expensive machinery for power conversion and storage. Additionally, the variability in wind speed and direction can limit its efficiency. Hydropower generates electricity by converting water flow through hydraulic turbines connected to generators. However, it is only feasible in locations with suitable water dams, and constructing large dams can lead to significant ecological and geological issues. Tidal energy faces similar challenges with conversion methods and availability, further complicating its widespread use. Driven by human curiosity and technological advancements, we have explored various energy sources and ultimately identified hydrogen as a promising option. Hydrogen is an ideal medium for energy storage and transport because it can be readily obtained from water and biomass. It produces no harmful emissions, making it an attractive fuel for the future [2]. Hydrogen energy is considered one of the most appealing clean energy sources due to its environmental benefits and high energy capacity [14]. Despite its abundance, hydrogen poses challenges in usage because of its flammable nature and the complexities involved in handling it safely. Hydrogen is a clean form of fuel with a high energy density per mass (120 MJ/kg) compared to other chemical fuels. The heating value of hydrogen is 141.8 MJ/kg at 298 K, which is higher than most fuels [9,15]. However, its widespread use is limited for two main reasons. First, hydrogen is primarily an energy carrier rather than a freely available resource. It is typically extracted from water or hydrocarbons, which involves complex separation processes. Second, transportation is challenging because hydrogen is gaseous at low temperatures. Common storage systems use high-pressure gas cylinders at pressures around 20 MPa, but handling and storing hydrogen can be tedious and dangerous. The liquid form of hydrogen also presents difficulties, as it requires cryogenic tanks maintained at low temperatures below 50 K. Current hydrogen production methods from fossil fuels are energy-intensive due to endothermic reactions [16,17]. Moreover, these conventional methods are not renewable and generate carbon dioxide. Just as we utilize solar energy stored in the earth’s crust in the form of crude oil, natural gas, and coal, solar energy can also be harnessed to produce hydrogen from water, offering a sustainable energy solution.

## 3. Hydrogen Production Techniques

Although hydrogen is abundant in the environment, it is not freely available. It is primarily found in water and biomass, which require separation through variable chemical processes or techniques. Hydrogen can be produced from natural gases using high temperatures in the presence of catalysts. Hydrogen is generated from a variety of materials such as hydrocarbons and nonhydrocarbons, including water-based biomass using photo, thermal, electric, chemical, bio-energies, and their combinations [15,18]. The major classifications of hydrogen productions techniques are displayed in Figure 1.

The methods used to extract hydrogen depend on the source material. Several techniques are employed to produce hydrogen from water or organic waste, including natural gas reforming, biomass gasification, biomass-derived liquid reforming, thermochemical processes, electrolytic processes, direct solar water splitting, PEC processes, biological processes, photobiological processes, and microbial biomass conversion [19,20,21]. Among these, natural gas reforming is commonly used worldwide to produce hydrogen from methane and similar natural gases [22]. Reforming gasoline, kerosene, etc., for hydrogen production is more attractive due to its efficient yield. If they are reformed using solar energy, the emission of toxic compounds can be avoided [23]. In biomass gasification, the biomass or any organic residues are converted into hydrogen using heat, steam, and oxygen without combustion [24]. Biomass-derived liquid reforming involves reforming liquids such as ethanol and other bio-oils from biomass to produce hydrogen, similar to natural gas reforming [25]. Thermochemical processes use heat to induce chemical reactions that release hydrogen from water and organic residues [26]. Biological methods for hydrogen production utilize microbes like bacteria and microalgae. In the photobiological process, microorganisms convert organic residues into hydrogen using sunlight [27]. Green microalgae or cyanobacteria can split water into oxygen and hydrogen using sunlight [28]. Some photosynthetic microbes can also break down organic compounds to produce hydrogen through photofermentative processes [29]. Additionally, “dark fermentation” involves microorganisms digesting biomass and releasing hydrogen without light [30]. Microbial electrolysis cells use electric current to produce hydrogen by applying energy and protons generated by microbes breaking down organic matter [31]. Micro-sized microbial fuel cells can produce hydrogen from human saliva for on-chip applications [32]. Hydrogen can also be generated from wastewater containing both bacteria and organic materials [32]. The types of hydrogen produced through these methods are classified differently depending on the associated byproducts and processes [33,34]. They are categorized as different types with color codes from green to pink depending on quality and purity, based on the production routes [15]. However, they do not have any globally standardized properties or codes. Even the color variations cannot be visibly differentiated. However, to differentiate the purity and type of generation process, they are named with different colors. Hydrogen generated from water using clean electricity from solar or wind power without CO_2_ emission is known as green hydrogen. The hydrogen produced mainly from natural gas using the steam reforming process with low levels of CO_2_ emission is known as blue hydrogen, and involves the capturing and storing of the co-generated CO_2_. Grey hydrogen is produced from natural gas or methane using the steam methane reformation process without separation of greenhouse gases. It is almost equal to blue hydrogen. If hydrogen gas is produced from black coal or lignite without any carbon capture process, it is called black and brown hydrogen. It can damage the environment due to toxic emissions. Hydrogen produced from fossil fuels by gasification is also sometimes called black and brown hydrogen. Hydrogen generated through electrolysis using nuclear energy is termed pink hydrogen. It is also called purple or red hydrogen. Turquoise or cyan hydrogen is produced by methane pyrolysis with solid carbon production. Hydrogen made through electrolysis using solar power is also called yellow hydrogen. Naturally available hydrogen from underground deposits and rocks formed by natural processes is called white hydrogen.

Among the different methods used to produce hydrogen, PEC hydrogen generation is particularly clean and environmentally friendly [35,36]. It uses only solar energy with semiconducting materials, making it an attractive option due to its low cost, abundance, and clean nature [37,38]. Compared to thermocatalytic processes, PC water splitting, which involves a photocatalyst that produces excited states through photon absorption, is more efficient. The principles of PC water splitting are detailed in the report by Hisatomi et al. [39].

## 4. Solar Hydrogen from Water Splitting

Water can be split into hydrogen and oxygen using either thermal or electrical energy through processes known as thermolysis or electrolysis, respectively [5]. Thermolysis involves splitting water using thermal energy [40], whereas electrolysis involves splitting water into hydrogen and oxygen by applying an electrical current from external sources [41]. These water-splitting methods are generally considered expensive due to the high energy requirements. However, using solar power directly can reduce costs. When solar power is used to generate hydrogen from water, the process is known as solar hydrogen production. This method involves direct solar water splitting, also referred to as the photolytic process, where solar light, along with catalytic material, is used to split water into hydrogen and oxygen under sunlight. The PEC process converts solar light into electrical energy using a semiconductor, which is then used to split the water. Researchers worldwide are exploring ways to generate hydrogen from water using solar energy without relying on external power sources. PEC water splitting combines solar and electrochemical cells to generate current from sunlight and use it for water splitting. This photoelectrochemical process is an ideal, environmentally friendly, low-cost, and renewable method for hydrogen production. The energy required for splitting water is supplied by the semiconductor/electrolyte junction incorporated within the PEC cell. The semiconducting materials should be chemically stable with an optical bandgap between 1.8 and 2.5 eV, suitable for absorbing white light. Solar hydrogen production via PEC water splitting has garnered significant attraction due to its potential for clean and economic hydrogen generation. For example, Penconi et al. studied the use of rare earth, gallium, and indium oxides as catalysts for water splitting using photolysis, sonolysis, and sonophotolysis processes. Their research compared hydrogen production with and without sulfur doping, and found that sonophotolysis, a hybrid of light and ultrasound, significantly improved hydrogen production [42].

### 4.1. Mechanism of Solar Water Splitting

Hydrogen can be generated through a simple water electrolysis process by applying a potential between electrodes immersed in water. There are three primary methods to split water into hydrogen and oxygen: PC, PEC, and photovoltaic–electrochemical (PV–EC) hybrid methods (Figure 2). [43]. In the PC process, a PC material is dispersed in a water reservoir and exposed to solar light. Typically, when a semiconductor with a bandgap between 1.3 and 3.1 eV is illuminated by visible light, electrons in the valence band (VB) are excited to the conduction band (CB). This excitation leads to the reduction of H^+^ to form hydrogen gas, while a hole remains in the VB [44,45]. This method is the simplest and most cost-effective among the three, but suffers from low solar-to-hydrogen (STH) efficiency and poses safety concerns for large-scale production due to open-space generation and difficulties in separating H_2_ and O_2_ [46].

The efficiency of solar hydrogen production by water splitting is termed solar-to-hydrogen (STH), and it is estimated using Equation (1) [2].
(1)STH=  [(jsc(mAcm2))×(1.23 V)×ηFPtotal(mWcm2)]AM1.5G
where *P_total_* is the power density of incident sunlight (AM1.5G), *j_sc_* is short-circuit photocurrent density, 1.23 V is the voltage required for water splitting, and $\eta_{F}$ is the faradic efficiency. The AM1.5 G is the global standard for light sources, with its power equal to 1000 W/m^2^ or 100 mW/cm^2^.

The solar-to-hydrogen conversion efficiency of a process can also be given as Equation (2), based on the amount of hydrogen generated.
(2)STH=[( mmole H2s)×237 kJ/molePtotal(mWcm2)×Area (cm2)]AM1.5G

The capacity of photocatalytic splitting of water to hydrogen (H_2_) is also expressed by apparent quantum yield as Equation (3).
(3)$$A.Q.Y=\frac{\begin{array}{c} number\;of\;evolved\;H_{2}\;molecules\times 2\;\\ \left(number\;of\;reacted\;electrons\right) \end{array}}{number\;of\;incident\;photons}\times 100\%$$

In the PEC process, sunlight is focused on an electrochemical system that includes a photoanode and a cathode submerged in an electrolytic water bath. Hydrogen and oxygen produced from the water under sunlight irradiation are directed to their respective electrodes, with an external potential applied to facilitate the collection of the products. Specialized semiconducting materials are used to improve efficiency. However, this method requires high costs for reactor fabrication with large-area electrodes and the electric potential needed to drive the water-splitting reaction. These semiconducting materials convert solar energy similar to PV solar cells, but are immersed in a water-based electrolyte bath.

In a PV–EC hybrid system, photovoltaic solar cells and electrocatalytic water-splitting cells are integrated into a single unit to produce hydrogen using solar energy without the need for additional power sources.

A photovoltaic cell coupled with a water electrolyzer can achieve efficient water splitting. When the incoming light energy exceeds the semiconductor’s bandgap, electron-hole pairs are generated within the semiconductor (Figure 3). This occurs as electrons are excited from the VB to the CB, leaving behind holes. The charge carriers, such as electrons and holes, diffuse to the semiconductor–electrolyte interface, where they drive redox reactions with available electron acceptors and donors. Photoexcited electrons reduce protons (H^+^) to hydrogen gas (H_2_) while water is oxidized to produce oxygen, as shown below [18,21,47].
2H_2_O + 4h^+^ ––> O_2_ + 4H^+^ (oxidation process)
2H^+^ + 2e^+^ ––> H_2_ (reduction process)

Typically, hydrogen generation and water splitting are distinct processes. In water splitting, single or many combined catalysts are used. In the single catalytic process (Figure 3a), water is split using visible light of energy higher than the bandgap of the catalytic material [48]. Over time, this single photocatalyst is not efficient due to its corroding nature and poor electron-conducting properties. Instead, if two catalysts are used as main and co-catalysts (Figure 3b), efficiency can be improved. The two-catalytic system is inspired from natural photosynthesis of green plants called Z-schemes (Figure 3). Here, a wide range of visible light can be utilized to produce efficient water splitting. Each catalyst can be activated with efficient function for a single process as either the oxidation catalyst or reduction catalyst using a proper redox mediator, which can boost the water-splitting and electron-conducting processes. Separate electron scavengers and sacrificial alcohols are used to facilitate reduction and oxidation reactions. In contrast, water splitting utilizes only pure water as both the reactant and solvents, without additional reagents [48].

### 4.2. Advantages of Solar Hydrogen

Hydrogen production from water is highly desirable due to its clean and efficient nature compared to alternative methods. While this process is often considered expensive because of its energy requirements, it can become more cost-effective when powered by solar energy and with proper recycling of the catalysts. Hydrogen production using solar power is referred to as solar hydrogen. PC water splitting is actively pursued for hydrogen production because it efficiently utilizes solar energy to address environmental and energy challenges. Photocatalysts driven by visible light are primarily used for solar energy conversion. By mimicking the natural process of photosynthesis, photoelectrolysis of water offers a viable route to generate sustainable, low-cost, and green hydrogen. Solar hydrogen generation is akin to artificial photosynthesis, where photoactive materials convert solar energy into electrical energy to split water into hydrogen and oxygen. This process utilizes abundant natural resources, i.e., water and sunlight, under mild conditions, without producing any harmful byproducts. Photocatalysts can range from simple oxides to complex engineered composites to improve the water-splitting process. For efficient hydrogen production, semiconducting materials are often employed as photocatalysts. These photocatalytic materials can be optimized for effective light harvesting and charge separation. Techniques such as bandgap engineering, morphology adjustments, and composite variations can enhance their performance. In addition to pure water, seawater, wastewater, and other biomass-based water wastes can also be used to generate hydrogen with solar energy. This advancement in solar-driven hydrogen production technology brings us closer to a sustainable energy future. Recent developments in this field are promising, indicating hydrogen from water could become a key alternative for meeting global energy demands while supporting sustainable development.

### 4.3. Materials and Methods for Solar Hydrogen Generation

Among the various methods for water splitting, PEC water splitting stands out for its efficiency and sustainability [49]. This method involves using specialized semiconducting materials to split water into hydrogen and oxygen under sunlight irradiation. The key steps in PEC water splitting are (i) generation of hole-electron charge carriers through absorption of light by semiconductors; (ii) separation and transfer of these charges to the semiconductor surface; (iii) interfacial reactions to split water into hydrogen and oxygen; and (iv) collection of the generated hydrogen. In this photoelectrochemical splitting of water, catalytic systems play major roles. The choice of catalytic material and its structural framework significantly affects the efficiency of the water-splitting process and the high-density yield. There are four catalytic reactions produced in the water-splitting process, namely, (a) photocatalysis, (b) electrocatalysis, (c) photoelectrochemical, and (d) photoelectrocatalysis, as explained by Li et al. (Figure 4) [50]. In photocatalysis, optical energy is used to trigger photoexcitation and produce hole-electron charges to split water (Figure 4a). When the energy required for water splitting is applied from an external energy source, it is called electrocatalysis (Figure 4b). In photoelectrochemical systems, photogenerated voltage is used to induce electrolysis of water (Figure 4c). In this self-driven system, the semiconductor electrode coated with electrocatalysts is the main factor for electrochemical conversion and storage. In photoelectrocatalysis systems, a photocatalyst and electrocatalyst integrated electrode is used to increase water-splitting efficiency (Figure 4d). This photoelectrocatalytic system efficiently uses sunlight to excite carriers and efficiently transfers the carriers to produce an efficient redox process.

Therefore, developing highly efficient materials that ensure stable, active photoconversion and catalytic reactions is crucial for effective solar hydrogen production. Co-catalysts are often used to enhance photoconversion and catalytic processes. These co-catalysts can offer multiple active sites to promote robust water-splitting reactions. Typically, noble metals are used as co-catalysts, but they are expensive and rare. Consequently, there is a strong need for abundant and cost-effective co-catalysts. Metal sulfides, metal oxides, metal oxynitrides, and some metal-free semiconductors have been extensively explored for solar water splitting. Since Fujishima and Honda’s pioneering work on hydrogen fuel using a TiO_2_ electrode for PEC water splitting, numerous advancements have been made in this field. A significant body of research focuses on semiconductor materials and their effectiveness in splitting water into hydrogen and oxygen under UV or visible light. The effectiveness of these materials depends on their response to the wavelength and energy of light [51]. For efficient STH reactions, catalytic materials must have an optimal bandgap for efficient PC reactions. A report by Hisatomi et al. consolidates the general issues and remedies associated with PEC water splitting using semiconductor photocatalysts [52]. Additionally, Maeda et al. discussed the effects of various semiconducting catalysts and co-catalysts on efficient PC water splitting [53]. Wang et al. developed a machine learning (ML) model to select efficient doping materials for improved water-splitting performance, and this model was validated by experimental results [54].

Metal oxide (MO) semiconductors are excellent materials for water splitting due to their unique properties. Various semiconducting MOs, such as ZnO, TiO_2_, CuO, Cu_2_O, MgO, and NiO, are used in PEC cells as photoelectrodes. Amongst these, cuprous oxide (Cu_2_O) stands out for its earth abundance, non-toxic, low cost, and high electrical conductivity properties. It is particularly well-suited as a photocathode for water splitting because it has a more negative potential compared to hydrogen evolution potential. Additionally, as a p-type semiconductor, Cu_2_O generates electrons at the electrolyte–semiconductor interface when used as a cathode in PEC water splitting, offering several advantages over n-type cathode materials [55]. Ibrahim et al. utilized pure and Ni-doped Cu_2_O films for hydrogen production via PEC water splitting under light illumination, demonstrating improved performance due to doping [55]. TiO_2_ has also been extensively explored as a photoelectrode for PEC water splitting since its initial study by Fujishima, and its combinations with other materials continue to be used for effective solar water splitting [56].

Solid solutions, such as CdS-ZnS, can be employed without any co-catalyst to produce hydrogen from water and biomass [57]. Yu et al. developed a multifunctional coral-like p-MoS_2_/NiS_2_ nanostructured electrocatalyst, which shows enhanced water-splitting performance [58]. Patil et al. reported a porous heterojunction of Zn_1−_*_x_*Cd*_x_*Se/ZnO nanorods (NRs) synthesized on zinc foil via a hydrothermal technique, demonstrating significant photocurrent generation useful for solar hydrogen generation [59]. Song et al. proposed a materials system for unassisted solar water splitting using multijunction tandem photoelectrodes as a sustainable approach [60]. Zhang et al. introduced a novel complex perovskite oxide structure for solar thermochemical hydrogen generation, with its properties studied using density functional theory (DFT)-based parallel Monte Carlo computations [61]. Fehr et al. discussed the technoeconomic feasibility of halide perovskite photoabsorbers for producing green hydrogen at <USD 2/kg using PEC cells [62].

Similarly, nitride materials are also promising for water-splitting reactions. Mehtab et al. reported a hydrogen generation rate of 467.2 μmol h^−1^ g-cat^−1^ for g-C_3_N_4_ nanosheets, which is higher than that of bulk g-C_3_N_4_, due to their efficient utilization of solar radiation. With an optical bandgap between 2.7 and 2.9 eV, g-C_3_N_4_ converts a significant portion of the solar spectrum into useful energy [63]. Metal nanoparticles dispersed on graphitic carbon nitride (g-C_3_N_4_) (M/CN) further enhance PC hydrogen yield. This improvement is attributed to the active sites created on the surface of g-C_3_N_4_ by the dispersed metal nanoparticles, which act as co-catalysts. Rawool et al. investigated the effects of different metal nanoparticles on g-C_3_N_4_ and observed variations in their characteristics [64]. They found that among different metals like Pt, Pd, Au, Ag, and Cu, Pt nanoparticles produced the best performance under both UV and sunlight. Nanoscale materials play a crucial role in both PC and PEC water-splitting processes [65]. In these nanoscale designs, graphene, transition metal dichalcogenides (TMDs), and g-C_3_N_4_ are key players in hydrogen production through PEC reactions. The bandgap and heterojunction formation in these 2D materials can be tailored to enhance light harvesting, charge separation, and stability in the process.

To enhance STH conversion, dye-sensitized co-catalytic materials are also employed to provide additional active sites [66]. Gupta et al. demonstrated an improved STH performance of 60 mmol h^−1^ g^−1^ under visible light irradiation using dye-sensitized 1T-MoSe_2_, which outperforms other semiconducting materials [67]. Yang et al. prepared an Eosin Y-sensitized structure with Rh_2_O_3_ nanoparticles on a reduced graphene oxide surface [68]. This structure exhibited a quantum efficiency of ~79.3% at 520 nm, surpassing that of Rh mixed with RGO catalysts. Carbon-coupled phosphates, metal selenides, and sulfides can also serve as co-catalysts, enhancing STH efficiency. Transition-metal nitrides (TMNs), known for their unique structural, electrical, and chemical stability, show promising results due to their co-catalytic properties. The negatively charged nitrogen atoms in TMNs impart a noble metal-like character comparable to Pt-based catalysts through modifications in d-band density. Chen et al. reported hydrogen production using Co_3_N/CdS nanocomposites under visible light, achieving better results than Pt/CdS. The H_2_ production rate for Pt/CdS has increased to ∼137.33 μmol h^−1^ mg^−1^ (λ > 420 nm), compared to ∼26.34 μmol h^−1^ mg^−1^ for pure CdS NRs [51]. TMN-based dye-sensitized systems are promising for STH conversion, though there are limited reports available on these materials. Their intrinsic properties can be tailored by doping with metal ions to improve PEC water-splitting efficiency. Liu et al. investigated various transition metals doped into cobalt nitride with Eosin-Y dye-sensitization for visible light-driven hydrogen production [66]. Their study showed efficient hydrogen production with low transfer resistance and high electrical conductivity. The adsorbed Eosin-Y effectively transfers photogenerated electrons to the catalytic materials’ surface, driving reactions to produce more hydrogen. Transitional metal borides (TMBs) are also effective substitutes for Pt as cocatalysts for visible light water splitting and H_2_ production [69]. These amorphous TMBs are abundant and cost-effective, potentially replacing expensive Pt co-catalysts. Ramirez et al. reviewed the water-splitting performance and related processes of rhenium-based materials [70].

Current electrocatalysts often require large cell voltages, which limits their efficiency. To address this, Zhu et al. developed a bifunctional electrocatalyst (SiO*x*/Ru) that activates both the hydrogen evolution reaction (HER) and the oxygen evolution reaction (OER), offering excellent performance for industrial-scale renewable energy conversion [71]. Instead of using inorganic semiconductors, organic semiconductors could eliminate the need for solar cells, as organic PEC cells can operate without potential biasing due to band position engineering. Additionally, organic electrodes can function as both photocathodes and photoanodes, depending on the order of material coatings. Cho et al. fabricated an organic photoactive catalyst composed of nickel-iron layered double hydroxides for the OER using photogenerated holes [72]. Plasmonic nanoparticles, used as photoelectrodes, can enhance photoactivity and improve hydrogen generation efficiency. These plasmonic particles increase optical absorption across a broad spectrum through multiple absorption and scattering behaviors. Chen et al. demonstrated a 200% enhancement in PC hydrogen generation using plasmonic resonance-induced photoelectrodes made of zinc nanorods and silver particle composite films [73]. Silver (Ag) plasmonic nanostructures embedded in zinc oxide (ZnO) nanorod arrays produced enhanced solar hydrogen production [73]. These Ag nanoparticles acted as nanoantennas, similar to gold (Au) nanoparticles, to generate plasmonic resonance and increase absorption [74]. Plasmonic nanostructures can boost photoelectrocatalysis due to their improved light-absorbing ability. Liu et al. designed a plasmonic nanostructure using Bi_3_(Se_n_Te_1-n_)_2_ ternary alloy nanowires for efficient hydrogen generation via solar water splitting [75].

Kotkondawar et al. reported the preparation and evaluation of a Pt/Au/CdS hetero nanostructure designed for PC hydrogen production. This structure incorporates a sulfide shuttle as a sacrificial agent to facilitate rapid hole-scavenging processes. The unique architecture of this electrode enhances its catalytic activity, including improved hole scavenging and charge transport properties. The multicomponent design, which features plasmonic nanoparticles, significantly boosts light harvesting, water splitting, and charge carrier collection efficiencies [76]. While platinum-based noble metals are commonly used as catalysts for solar hydrogen generation, their cyclic stability is often limited. Consequently, non-noble metal-based materials, such as oxides, carbides, sulfides, phosphides, and selenides, are being explored as effective heterogeneous electrocatalysts for solar hydrogen generation. Additionally, carbon-based materials are gaining attention due to their unique physicochemical properties. Joshi et al. prepared carbon fiber-based materials coated with nano molybdenum disulfide (MoS_2_) for solar hydrogen production through membrane-less electrochemical water splitting [77]. They achieved an STH conversion efficiency of 2.46% at 35 °C with 430 W/m^2^ irradiation. The PEC performance of a system largely depends on its ability to absorb sunlight and utilize photoexcited carriers for water splitting. Many systems struggle with these aspects. To address these efficiency issues, Tian et al. proposed a design to enhance the overall performance of photoassisted water splitting [50]. This design integrates charge transfer channels monolithically, using a combination of carbon dots/carbon nitride nanotubes and FeOOH/FeCo layered double hydroxide nanosheets. The effectiveness of this design has been verified both experimentally and theoretically through DFT modeling.

The development of high-efficiency STH conversion systems remains a significant challenge. Poor light absorption and carrier recombination degrade the development performance of STH systems. Photogenerated holes and electrons can recombine, reducing the efficiency of the PC mechanism. Recombination is more likely to occur on the way to the surface, with defects, grain boundaries, and ion vacancies in semiconductors acting as recombination centers. To address this, specially designed structural catalysts are employed to minimize recombination and enhance water-splitting reactions. Wang et al. reported that the PtTe_2_/Sb_2_S_3_ nanoscale heterostructure is a promising photocatalyst for efficient water splitting. This heterostructure operates as a direct Z-scheme system for solar water splitting [78]. To improve charge transport in PEC water splitting, You et al. introduced a certain amount of W^5+^ into Bi_2_WO_6_ crystals to enhance their photochemical properties [79]. The induced internal electric field improves charge separation and enhances the piezoelectric photoelectrochemical (piezo-PEC) water-splitting process. Some semiconductors are not active across the full solar spectrum, limiting their water-splitting efficiency. To address this, Park et al. designed an active photoanode by linking ligands with semiconductors to enhance water-splitting and charge-collecting performance [80]. The morphology of the catalyst also plays a crucial role in PC hydrogen production. Specific crystal facets can be more effective than others. Liu et al. demonstrated that CeO_2_ nanorod (311) facets exhibit higher PC activity compared to CeO_2_ (220) facets [81]. Yu et al. used facet engineering to modify Bi_2_YO_4_Cl catalytic material, forming plate-like particles that expose specific crystal faces and improve catalytic performance for solar hydrogen generation [58]. Pihosh et al. prepared polycrystalline tantalum nitride (Ta_3_N_5_) as photoanodes for solar hydrogen production [82], showing high photocurrent generation. Silicon, a widely used material in solar cells, can also be used for solar water splitting. Chandrasekaran et al. reviewed various silicon nanostructures and their impacts on solar water electrolysis [83]. They fabricated highly efficient multishell ITO/WO_3_/BiVO_4_/CoPi NT electrodes for PEC water splitting using a soft-template methodology, which increased the active surface area and enhanced electrochemical activity [84]. Ti_3_C_2_ MXene composites are extensively studied for PC hydrogen production due to their large surface area and ease of functionalization. However, weak interactions at the Ti_3_C_2_ interface can impair carrier transport. To overcome this issue, Wei et al. proposed a sandwich structure of Ti_3_C_2_/R-TiO_2_ [58]. Liu et al. developed two imine-based 2D covalent organic frameworks (COFs), modified with benzotrithiophene moieties, to enhance catalytic activities for hydrogen generation through improved optical bandgap, crystallinity, and porosity [85]. Gao et al. designed two COFs, TPE-TPB-A and TPE-TPB-B, and studied their efficiency in PC hydrogen evolution reactions, reporting improved hydrophilicity and better charge separation and transport [68]. Among these, TPE-TPB-B showed superior performance. Direct seawater electrolysis is challenging due to side reactions and electrode degradation caused by impurities. To address this problem, Zhang et al. developed a direct electrochemical seawater splitting device [86]. This method uses only electricity and air: first, seawater is desalinated through oxygen reduction, removing CO_2_, Ca, and Mg ions in the formed alkaline environment. The softened seawater then forms artificial reefs, and is used for hydrogen production.

The efficiency of photocatalysts for water splitting often suffers from high overpotential, charge recombination, and poor surface redox activity. To address these issues, various technical and scientific strategies can be employed. Ding et al. reviewed different techniques and catalytic systems aimed at enhancing PC electrochemical efficiency for solar hydrogen generation [87]. One major factor contributing to reduced electrode lifetime and instability of cocatalysts in photoelectrochemical devices is photocorrosion. This effect can be mitigated by using stabilizing coatings to protect electrodes. Wang et al. improved the stability of the BiVO_4_ photoanode by applying a transparent CoFe-dispersed polyacrylamide (PAM) hydrogel coating [85]. The notable results of solar water splitting are consolidated in Table 1.

## 5. Hydrogen Storage Mechanisms

Although hydrogen is abundant in water and biomass, it is not freely available in a usable form. When utilized, it produces only water as a byproduct. However, its adoption has been limited due to the lack of a safe and economical storage system [91]. Additionally, its intermittent nature, due to the reliance on solar light, necessitates the use of additional storage devices. Figure 5 shows the different methods available for hydrogen storage. The conventional hydrogen storage methods such as gaseous or liquid forms present safety and practical challenges. Gaseous hydrogen requires high-pressure storage tanks, while liquid hydrogen needs extremely low temperatures of around –250 °C, requiring expensive vessels. Moreover, storing hydrogen in gaseous or liquid form is not practical for office environments or electronic and computing facilities [92,93]. In contrast, storing hydrogen in a solid form, such as within catalytic electrode materials, allows for easier use in portable batteries and miniature fuel cells, as illustrated in Figure 6.

### Solid Storage of Hydrogen

Solar hydrogen is considered a promising alternative among renewable energy sources due to its several advantages, including being environmentally friendly with no carbon emission, non-toxic emission, having high energy density, and being sustainable [94]. Storing this hydrogen in solid form simplifies transport and recovery, overcoming the issues associated with liquid or gaseous hydrogen storage methods. Hydrogen can be stored in solid form using various materials and processes, including (1) hydrogen adsorption onto high-surface-area materials, (2) hydrogen inclusion in interstitial sites of host materials, (3) chemical bonding of hydrogen with covalent and ionic compounds, and (4) oxidation reactions with metal ions. These materials are mainly classified into three categories based on the chemical process of storage: (i) dissociative, (ii) chemically bound, and (iii) adsorption materials. In the dissociative process, molecular hydrogen is dissociated into hydrogen atoms, which then occupy interstitial sites within the material. Alternatively, hydrogen atoms or molecules can chemically bond with material. In the case of adsorption, molecular hydrogen attaches to the material’s surface through weak van der Waals forces, a process also known as physisorption [11]. Chemical absorption of hydrogen by materials is considered a promising method for solid hydrogen storage due to its high storage capacity and ease of transportation.

However, solid-state storage methods, such as chemisorption (forming metal and chemical hydrides) and physisorption, face challenges such as high thermodynamic requirements, slow kinetics, and poor reversibility. This has driven the scientific community to search for more efficient solid-state hydrogen storage media. Hydrogen is typically stored in solid form either as hydrides or hydrogen molecules, depending on the storage materials and methods used. Effective hydrogen storage materials should possess good gravimetric and adsorption properties and low adsorption energy, allowing for easy desorption with minimal energy expenditure. Hydrides often require high temperatures for desorption, which can be problematic, while hydrates need high pressure to form. Various materials, such as metal oxides, metal hydrides, porous carbon, and their combinations, are being explored to identify efficient hydrogen storage solutions, yielding a range of results.

## 6. Materials for Solid Hydrogen Storage

Storing hydrogen in solid form poses numerous challenges related to materials and methods [95]. Hydrogen is an efficient energy source and has the potential to address many challenges in the quest for sustainable energy. However, handling liquid and gaseous hydrogen for energy extraction can be difficult. Fortunately, solid-state hydrogen storage is more manageable in terms of storage and transportation. Key factors in solid hydrogen storage include storage density, dehydrogenation temperature, and the dynamics involved. According to the U.S. Department of Energy, a gravimetric capacity of 6.5 wt% is required within the temperature range of −40 to 60 °C for an optimal energy strategy [11]. Current theoretical densities of available storage materials fall short of this requirement, making it challenging to produce efficient solid storage materials at room temperature. The storage capacity of these materials depends on their properties and their chemical interactions with hydrogen. Enhancements in material structures and composite properties can lead to more efficient storage materials.

### 6.1. Experimental Works on Solid Hydrogen Storage

There are several ongoing investigations into novel materials and methods for improving hydrogen storage efficiency. The surface chemical interactions of functionalized nanoparticles in composite materials are influenced by their crystal facets, which affect hydrogen adsorption and storage capacity. For example, Dun et al. prepared Mg particles on reduced graphene oxide with different facets, and studied their hydrogen storage properties. They found that Mg with a (2$\bar{11}$6) crystal surface exhibited increased hydrogen absorption, reaching up to 6.2 wt% [96].

Eno et al. found that systems decorated with Ag and Au demonstrated superior chemisorption behavior [97]. Graphdiyne is a new low-dimensional carbon material like graphyne; these are two-dimensional carbon allotropes of graphene with honeycomb structures [98]. Hydrogen doping with oxide materials can enhance the storage performance due to structural variations induced by adsorption and desorption. Hydrogen sorption is typically more effective in the bulk phase compared to surface adsorption. At high temperatures, hydrogen storage by adsorption improves because the bulk diffusion of hydrogen overcomes activation barriers more easily. Lee et al. investigated the hydrogen storage capabilities of electrochemically formed VO_2_ nanotubes [99], finding that high temperatures were necessary to achieve substantial hydrogen adsorption. Hydrogen molecules enter the lattice sites of metal oxides, affecting their electrical and structural properties.

Metals, intermetallic compounds, and alloys generally react with hydrogen to form metal–hydrogen compounds, or hydrides, at elevated temperatures. These hydrides can be ionic, covalent, volatile covalent, or metallic, depending on the nature of the metal and its interaction with hydrogen. Light metals from groups I, II, and III combine with hydrogen to form metal–hydrogen complexes, which can store hydrogen at high volumetric densities. Many metals and alloys can reversibly absorb large amounts of hydrogen. Electropositive metallic elements, such as yttrium, lanthanides, scandium, actinides, and members of the titanium and vanadium groups, are highly reactive with hydrogen and are thus suitable for hydrogen storage [3]. While metal hydrides are considered for solid hydrogen storage, their slow release rates, high temperature requirements, and low adsorption densities limit their practical use. Rare earth materials are also considered for hydrogen storage, but their low storage density and high cost hinder their application. Titanium and iron-based materials offer a lower-cost alternative with low operating temperatures, though their storage density remains low.

For storing hydrogen in solid form, solid-state hydride materials are anticipated to play a crucial role in developing safe, energy-efficient, and high-energy-density systems. To enhance the hydrogen storage capabilities of metal hydrides, a nanostructuring strategy has been employed. Cho et al. prepared nanostructured hydrides by incorporating metallic nanoparticles with nanostructured scaffolds, creating nanoconfinements that provide numerous nanointerfaces suitable for advanced hydrogen storage [100]. Chen et al. summarized solid-state hydrogen storage properties of various hydride materials [101]. Similar to metal hydrides, metal borohydrides, which exhibit ionic properties, can store hydrogen at high gravimetric densities. Combining metal hydrides with metal borohydrides can enhance hydrogen adsorption.

Sodium borohydride (NaBH_4_) is a promising material for hydrogen storage, with its efficiency potentially enhanced through doping. Salman et al. doped NaBH_4_ with transition metals and studied their hydrogen storage abilities, showing good adsorption and release properties at reduced temperatures [102]. Transition metal hydrides also demonstrate improved performance in both hydrogen generation and storage. Berlouis et al. investigated nickel-ceria systems for hydrogen storage properties and observed variations depending on the preparation method [103]. Ding et al. prepared V-based solid solution alloys co-doped with Cr and the rare earth Y for hydrogen storage studies, finding that a Ti−V−Mn−Cr−Y alloy achieved hydrogen adsorption of ~3.71 wt% within 50 s at 6 MPa. The addition of rare earth elements to V improved the hydrogen storage performance of the composite [104]. Shashikala et al. studied Ce substitution in hydrogen absorption properties for the Ti–V–Fe series and found that Ce inclusion enhanced and improved hydrogen [105].

In solid-state hydrogen storage, both physisorption and chemisorption processes can be involved. Physisorption dominates in porous materials with high surface areas, while chemisorption is common in hydrides. Each process has its advantages and disadvantages depending on the materials, structures, and associated reactions. Xia et al. reviewed recent advancements in porous carbon-based materials and their benefits for solid hydrogen storage [106]. Nanostructured materials, due to their inherent porosity and high surface area, are extensively explored for hydrogen storage and generation. Materials such as carbon nanotubes, graphene, boron nitride nanotubes, and metal-organic frameworks (MOFs) are commonly used for hydrogen storage through either physisorption or chemisorption. Carbon-based materials are particularly attractive for hydrogen storage because of their high surface area-to-volume ratio and ease of preparation. It is important to formulate material design strategies to have innovative kinetic and thermodynamic properties for efficient storage properties. Nanostructuring and doping can also reduce kinetic barriers in complex hydrides. The incorporation of nanostructured metal hydrides into carbon scaffolds, as shown in Figure 7 [100], can strategically boost interfacial properties to produce efficient catalytic activities by enhancing functional properties.

For instance, Stock et al. prepared nanoporous carbon from coffee waste and found it achieved fully reversible storage with a gravimetric density of 5.79 wt% at 37 bar under cryogenic temperature (77 K) [107]. They carbonized coffee waste under nitrogen gas flow, with chemical activation using solid potassium hydroxide to produce a high surface area of up to 3300 m^2^/g. Among carbon-based materials, C_60_ fullerene is highly appealing for hydrogen storage. Buckyballs can store up to 8% of their weight at room temperature, surpassing the target of 6%. Researchers at Rice University have demonstrated that tiny C_60_ buckyballs can hold a high volume of hydrogen gas at room temperature. Additionally, transition metal atoms bound to fullerenes and organometallic buckyballs can adsorb a high density of hydrogen atoms under ambient conditions [108,109].

A highly aligned graphene oxide multiwalled carbon nanotube (MWCNT) composite has shown a high hydrogen storage capacity of 2.6 wt% at room temperature. Aboutalebi et al. used a liquid crystal route to prepare a 3D framework of graphene oxide layers, creating a large interlayer space for more hydrogen storage. The inclusion of MWCNTs as 1D spacers within the graphene oxide further enhanced hydrogen storage. This approach is promising for the scaling-up of efficient hydrogen storage applications [110]. MOFs are highly nanoporous due to the linkage of metal ions with organic ligands. MOFs are versatile, and their structures and surface areas can be controlled [108]. MOFs and zeolitic imidazolate frameworks (ZIFs) are known for their chemical, thermal, and mechanical stability. Rafael et al. demonstrated improved volumetric hydrogen storage capacity of ZIF-8 synthesized via mechanochemical methods, both as powder and pellets [111]. The notable experimental works available for solid hydrogen storage are listed in Table 2.

### 6.2. Theoretical Works on Solid Hydrogen Storage

Numerous new materials have been proposed for efficient hydrogen storage and release, though many of these are still under theoretical evaluation. There is a large number of works on solid storage of hydrogen, though majority of the works are in simulation studies, as shown in Table 3. Guangfen et al. investigated the hydrogen storage effectiveness of bare and metal-coated boron buckyballs (B_80_) using DFT. They identified Ca and Sc as the most effective metals for hydrogen storage in B_80_ composites. Their study demonstrated two types of adsorption mechanisms: a charge-induced dipole interaction and a Dewar–Kubas interaction [112]. Jaiswal et al. explored the hydrogen storage capacity of Ar-encapsulated Si_12_C_12_ heterofullerenes with Li functionalization using DFT simulations. They analyzed hydrogen molecule adsorption on Si_12_C_12_Li_6_, Ne@Si_12_C_12_Li_6_, and Ar@Si_12_C_12_Li_6_ structures, showing that the Li-functionalized Ar@Si_12_C_12_ cage exhibited the highest hydrogen adsorption energy [113]. Eno et al. examined the stability and reactivity of hydrogen interactions with Cu-, Ag-, and Au-decorated aluminum nanotubes using DFT. Jiang et al. reported on the hydrogen adsorption properties of NLi_4_-decorated graphdiyne nanosheets based on DFT models, noticing high gravimetric values of up to 8.91 wt% for various structures [114]. Brinkman et al. discussed the hydrogen storage and emission properties of various metal oxides and their mechanisms using theoretical modeling [91]. Morita et al. described a modeling-based study on the catalytic functions of Pt/Li_2_ZrO_3_/Pt and Pt/Li_4_SiO_4_/Pt metal oxide structures for water splitting and hydrogen storage using ion beam analysis techniques [115]. They observed up to 5 wt% hydrogen absorption in the Pt/Li_2_ZrO_3_/Pt sample at room temperature. However, Pt/Li_4_SiO_4_/Pt demonstrated better performance than Pt/Li_2_ZrO_3_/Pt. Niaz et al. studied a hydrogen storage system using a C_2_H_4_Nb complex with a DFT model [116], finding that its stability increased with the amount of adsorbed H_2_ molecules. Huang et al. summarized the properties of different metal borohydrides (LiBH_4_, NaBH_4_, and KBH_4_) for hydrogen storage using DFT simulations [11]. They also reviewed theoretical hydrogen storage densities of metal borohydrides, metal alanates, ammonia borane, metal amides, and amine metal borohydrides, noting improvements through possible method modifications [11]. Borospherene (B_40_), a new boron-based nanostructure similar to C_60_, has various applications. Bai et al. explored the potential of Li-decorated B_40_ for hydrogen storage using DFT calculations and found it to be an ideal complex for storing and releasing hydrogen, with each Li site capable of adsorbing up to six H_2_ molecules [117]. Estefania et al. studied the hydrogen adsorption effects of palladium clusters doped into graphdiyne and boron-graphdiyne using DFT calculations, reporting a reduction in hydrogen adsorption with palladium doping compared to pure materials [118]. Luis et al. investigated carbyne-decorated alkali, alkaline earth, and transition metals using the DFT method for hydrogen storage applications and found these materials to be highly efficient [119]. Compared to experimental works, there more theoretical works for solid hydrogen storage. This indicates that more exploration studies are required to find viable materials and methods.

### 6.3. Utilization of Solid Stored Hydrogen

In addition to production and storage, utilizing stored solid hydrogen is another critical aspect of hydrogen energy applications. Hydrogen fuel cells are particularly attractive for efficiently converting stored hydrogen into electricity, producing only water as a byproduct. Hydrogen produced from water via PEC water splitting can be easily converted to electricity using fuel cells. The solid oxide fuel cell (SOFC) is a highly efficient hydrogen utilization system, which can directly convert hydrogen into electrical energy with high efficiency and without pollutant emission. NaBH_4_ is a potential material for hydrogen storage and release, which is useful for fuel cell applications. Arzac et al. demonstrated a prototype fuel-cell kit coupling with a hydrogen generator for the hydrolysis of NaBH_4_ (Figure 8) [120].

This demo-kit is useful to understand the solid storage and use of hydrogen stored in a solid medium. This demo-kit consists of a water electrolyzer, gas storage tanks, fuel cell, and electric fan, and demonstrates the principle and full process of stored hydrogen application in an easy and economic manner.

## 7. Current Challenges and Prospects

The increased demand for improved quality of life necessitates greater use of natural resources. However, the Earth’s environment is struggling to cope with the effects of climatic change caused by the overexploitation of these resources. Hydrogen is increasingly seen as a promising solution to mitigate environmental pollution. Given the current state of the climate and its causes, there is growing interest in green hydrogen and its applications. The primary challenge lies in developing viable methods and materials for hydrogen generation and storage.

Adsorption material-based solid storage systems are identified as a highly efficient, low-cost, and safe method for hydrogen storage. In these material-based storage techniques, hydrogen is absorbed either as atoms or molecules on the adsorbent materials via physisorption or chemisorption. However, the US-DoE target of hydrogen storage capacity can be achieved only if the hydrogen is adsorbed in molecular form through quasi-binding mechanisms with low binding energies. For efficient reversibility of the solid storages systems, the H_2_ adsorption energy should be in between 0.1 and 1 eV/H_2_ at ambient condition, but earlier results have only produced this adsorption energy at elevated pressure and temperatures.

In the case of hydrogen generation, solar hydrogen generation from water is highly attractive; a large effort is currently taking place to attain the required levels, but good results have not yet been produced. As summarized in this report, several material systems have been developed over the past few decades for hydrogen generation using photo-induced water-splitting processes. However, no sustainable and feasible system has been established for this process. Different material systems, such as high surface area, doping, multi-composites, and dye-sensitized materials, have been effectively used to increase the efficiency of photocatalytic water splitting. However, the efficiency of this process not only depends on solar light and material systems, but it depends on many factors like photo-induced charge separation, utilization of a large fraction of the incident energy, prevention of recombination of created charges, and fast conduction of the produced charges. Hole scavengers or sacrificial agents are used to minimize the charge recombination process. These scavengers should also be applied continuously to obtain continuous hydrogen production. Tailoring the material properties requires utmost care in this direction. Functionalized layered materials can yield favorable outcomes.

Similarly, hydrogen storage also depends on the material systems, focusing more attention on materials-based research to find efficient H_2_ storages materials, which can store reversibly at high gravimetric density levels at ambient conditions, is required. Tuned nanointerfaces can bring additional kinetic changes for favorable hydrogen adsorption, improving storage efficiency. In addition, adsorption and desorption of H_2_ molecules should be reversible under ambient thermodynamic conditions to ensure efficient reversibility, safety, and cost-effectiveness. As far as metal hydrides are concerned, chemical reaction-based storage mechanisms are used. In complex hydrides, various reaction steps take place, slowing the H_2_ adsorption rate and causing low storage efficiency. Hence, these hydride materials alone cannot be used for practical applications. However, their combination with other nanostructured materials can improve their performance when producing nanointerfacial activities.

Although significant progress has been made, and some companies are actively using these methods, broader public adoption is still lacking. Currently, hydrogen technology is in its early stages, and further advancements are required to enhance production, storage, and conversion processes. More research is needed to discover potential new materials and methods for hydrogen production and storage. Once standardized methods and materials are established, the advantages of hydrogen are likely to attract public interest, quickly leading to increased action by government agencies. In the case of cost-wise perception, solar hydrogen generation appears to be costly. However, it is an energy-intensive process. Compared to other hydrogen generation methods, solar hydrogen generation using water-splitting techniques is a green and comparatively cheap method. If pure electrolysis of water is used, it is costly, as it requires a separate source of energy, whereas in solar water splitting, solar energy is utilized. The direct photocatalytic process is the most straightforward and economic process among the different methods. Understanding current materials and techniques will help in discovering more effective methods for hydrogen generation and storage. While there are some methods available for high-density storage, they have not yet achieved universal acceptance due to certain drawbacks. Therefore, it is crucial to explore alternative options that provide efficient energy sources without limitations. The initial cost may be high, but once established, it will become sustainable. Mainly, in the case of green hydrogen generation, renewable energies such as wind and solar are used, which can reduce the cost. According to the Hydrogen Council, the cost of green hydrogen production ranges from USD 3.5 to USD 7.5 per kilogram. Though it seems to be costly, it depends on many factors such as geographical location, transportation, and storage, among others. Solar hydrogen and solid storage will be viable techniques, considering their advantages. This combined strategy of solar hydrogen generation from water, solid storage, and utilization is more advantageous in all respects compared to other techniques. Many efforts are still taking place, which can reduce the cost through technological advancements, making it a more competitive alternative to fossil fuels. Apart from economic issues, it is highly beneficial for human society and its ecological, issues providing fully green energy without any harmful byproducts. Particularly, the use of hydrogen fuel cells for vehicles is more beneficial. The hydrogen storage capacity can also be further upgraded by fine-tuning nanomaterials using doping/decorating, functional coating, chemical activation, and creating vacancy/defect sites. Globally, several research labs and agencies like NREL [National Renewable Energy Laboratory, US] [121], HARC (Houston Advanced Research Center, USA), and IEEJ (The Institute of Energy Economics, Japan) are working on hydrogen production and storage. The International Renewable Energy Agency (IRENA) is also working to periodically review and consolidate the works available on hydrogen energy and suggest further improvements towards bringing it to full application [122,123,124]. This review aims to be a valuable resource for research communities by summarizing the current state of technologies and materials for hydrogen generation and storage.

## Figures and Tables

**Figure 1 nanomaterials-14-01560-f001:**
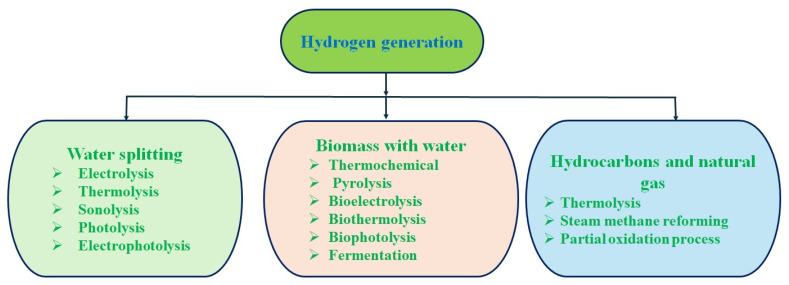
Hydrogen production techniques.

**Figure 2 nanomaterials-14-01560-f002:**
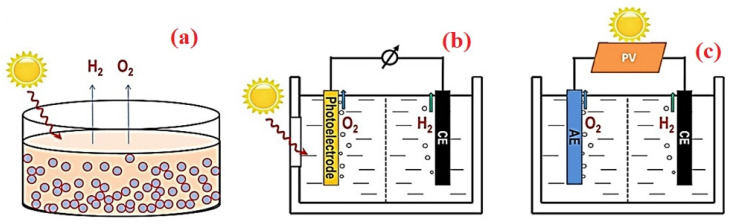
Schematic representations of (**a**) photocatalytic, (**b**) photoelectrocatalytic, and (**c**) photovoltaic–electrochemical water-splitting processes.

**Figure 3 nanomaterials-14-01560-f003:**
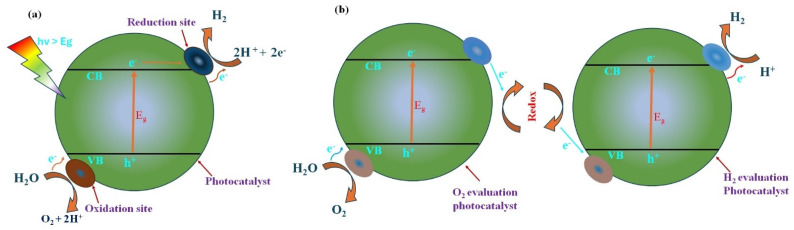
Schematic representation of the photocatalytic water-splitting process using (**a**) single and (**b**) double catalysts.

**Figure 4 nanomaterials-14-01560-f004:**
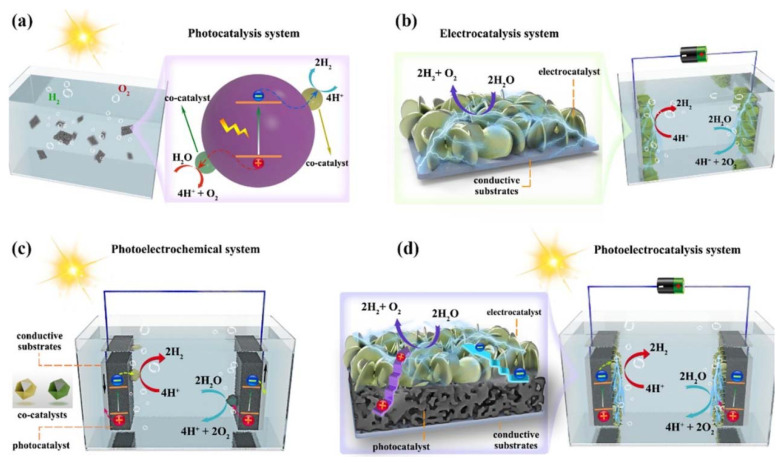
Four types of water-splitting systems that produce hydrogen and oxygen: (**a**) photocatalysis, (**b**) electrocatalysis, (**c**) photoelectrochemical, and (**d**) photoelectrocatalysis (reproduced with copyright permission) [50].

**Figure 5 nanomaterials-14-01560-f005:**
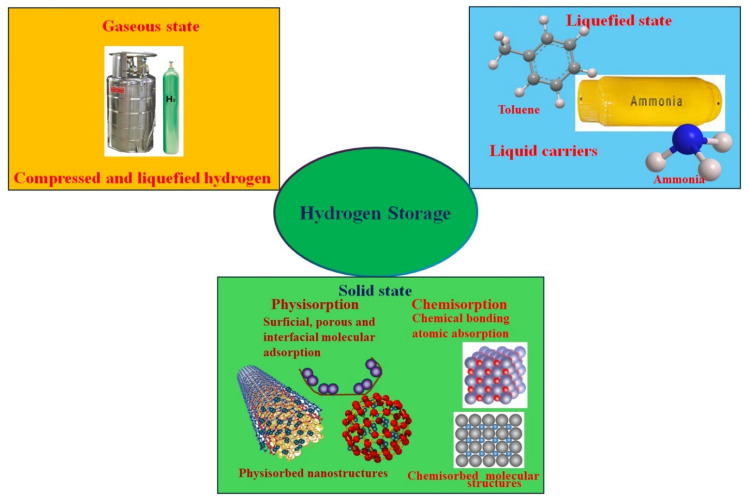
Different hydrogen storage methods.

**Figure 6 nanomaterials-14-01560-f006:**
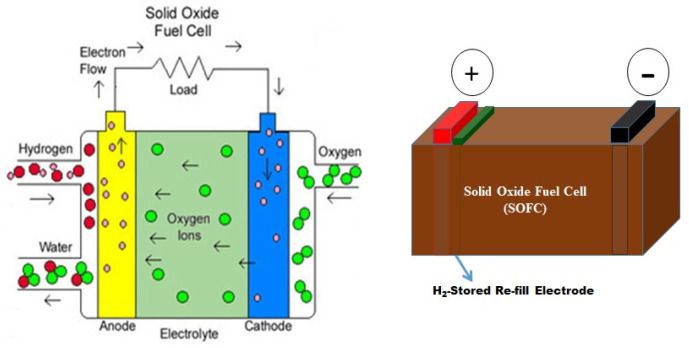
Schematic of solid oxide fuel cell’s process and structure.

**Figure 7 nanomaterials-14-01560-f007:**
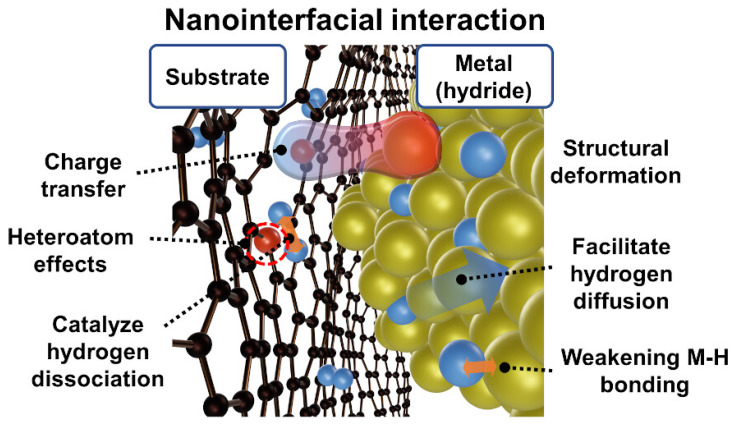
Schematic illustration of nanointerfacial interactions and the catalytic mechanism for hydrogen storage, which depicts possible methods of interaction and the resulting changes in hydrogen storage properties (reproduced with copyright permission) [100].

**Figure 8 nanomaterials-14-01560-f008:**
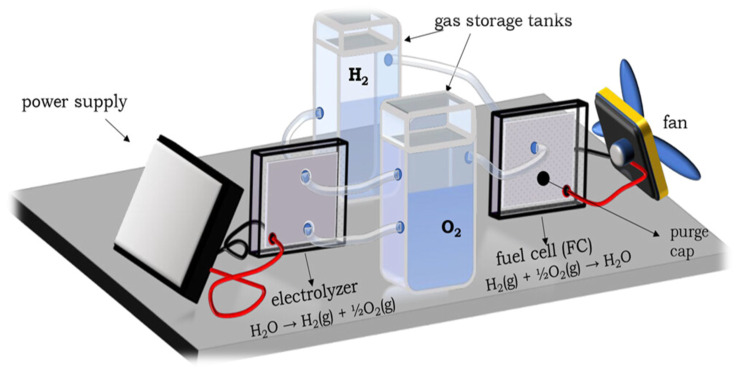
Prototype fuel cell kit coupling both the hydrogen generator and utilization of produced hydrogen for electrical energy generation [120].

**Table 1 nanomaterials-14-01560-t001:** Hydrogen generation efficiency of different materials using solar water splitting.

S. No	Material/ Method	Light Source	STH Conversion Efficiency (AQE)/ H_2_ Yield/Photocurrent Density	Ref
1	FACsPb(IBr)_3_ Perovskite	AM1.5G	photocurrent density = 12.5 mA cm^−2^ AQE = 15%	[60]
2	NiCoB/CdS	300 W Xe lamp	AQE = 97.42% H_2_ yield = 144.8 mmol h^−1^ g^−1^	[69]
3	Co_3_N onto CdS NRs	300 W Xe lamp	AQE = ∼14.9% H_2_ yield = 137.33 μmol h^−1^ mg^−1^	[51]
4	Rh/RGO	Visible light	AQE = 79.3% H_2_ yield = 98.1 mmol h^−1^ g^−1^	[68]
5	Thiol ligand-AgBiS2 NC	AM1.5G	AQE = 67%	[80]
6	CeO_2_/CdS	300 W Xe lamp	444 μmol g^−1^ h^−1^	[81]
7	Nano-g-C_3_N_4_/ Cu dendrites	AM1.5G	H_2_ yield = 59.2 μmol/cm^2^	[88]
8	Ni-doped Cu_2_O	400 W Xe lamp	photocurrent density = 5.72 mA cm^2^	[55]
9	Ti_3_C_2_/R-TiO_2_	300 W Xe lamp	H_2_ yield = 1.62 mmol g^−1^ h^−1^	[58]
10	V−CoN-Eosin-Y	-	AQE = 38% H_2_ yield = 21.21 μmol mg^−1^ h^−1^	[66]
11	Ta_3_N_5_ NRs	AM1.5G	photocurrent density = 10.96 mA cm^−2^	[82]
12	Bi3(Se_n_Te_1−n_)_2_	-	AQE = 22% photocurrent density = 13.8 mA cm^−2^	[75]
13	Pt/Au/CdS	Mercury lamp	AQE = 4.20% H_2_ yield = 15 mmol h^−1 ^g^−1^	[76]
14	Zn_1−_*_x_*Cd*_x_*Se/ ZnO NR	300 W XE lamp	H_2_ yield = 199 μmol cm^−2^/3 h photocurrent density = 7.8 mA cm^−2^	[59]
15	Bi_2_YO_4_Cl	AM 1.5G	AQE = 2.52% photocurrent density = ∼1.57 mA cm^−2^	[58]
16	LaSrMn/ FCoAlO_3_	-	H_2_ yield = 89.97 mmol h^−1^ g^−1^	[61]
17	[Fe(CN)_6_]_3_	-	photocurrent density = 320 mA cm^−2^	[89]
18	CdS/CdSe (QD)-ZnO NWs	-	AQE = ∼45% photocurrent density = ∼12 mA cm^−2^	[90]
19	NiFe-LDHs	AM1.5 G	AQE = 4.33% photocurrent density = 15.1 mA cm^−2^	[72]
20	Bi3(Se_n_Te_1−__n_)_2_ ternary alloy	-	AQE = 22% photocurrent density = 13.8 mA cm^−2^	[75]
21	CoFe-PAM/ BiVO4	AM 1.5G	photocurrent density = 5.7 mA cm^−2^	[85]

**Table 2 nanomaterials-14-01560-t002:** Hydrogen storage ability of different material systems.

S. No	Materials/Method	Storage Capacity	H_2_-Releasing Temperature	Ref
1	Pt/Li_2_ZrO_3_/Pt/	5 wt%	573 K	[111]
2	Vanadium doped-Sodium borohydride (NaBH_4_)/experimental	Releasing 5.3 mass% H_2_	355 °C	[102]
3	Aligned structures of GO–MWCNT composite	Up to 2.6 wt%	Room temperature	[110]
4	Ti−V−Mn−Cr−Y alloys	2.53 wt% hydrogen capacity	423 K	[104]
5	Ni/Ce composite	0.24 wt%	327 C	[103]

**Table 3 nanomaterials-14-01560-t003:** Theoretical works on solid hydrogen storage.

S. No	Materials/Method	Storage Capacity	H_2_-Releasing Temperature	Ref
1	Li atom-decorated Ar@Si_12_C_12_ cages/DFT Modeling	Gravimetric density of 9.7 wt%	100−120 K	[113]
2	Activated carbyne/ DFT Modeling	9 to 15 wt%	Near ambient conditions	[116]
3	NLi4-decorated graphdiyne nanosheets/ DFT Modeling	8.91 wt%,	-	[114]
4	Preferentially oriented Mg/rGO Hybrids	6.2 wt%	-	[96]
5	Lithium-decorated Borosphere Ti-B40-*n*H_2_	13.8 wt%	-	[117]
6	Cu-, Ag-, and Au-decorated aluminum nanotubes	5.8 wt%	-	[97]
7	Metal-coated B80 Buckyballs (Ca_12_B_80_)	9.0 wt%	-	[112]
8	Chemically activated carbyne (YC12-7H2)	9 to 15 wt%	Room temperature	[119]

## Data Availability

No new data were created or analyzed in this study.

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
