# Peer review of "Solar Hydrogen Production and Storage in Solid Form: Prospects for Materials and Methods"

_nanomaterials, 2024, doi:10.3390/nano14191560_

Round 1

Reviewer 1 Report

Comments and Suggestions for Authors

This review article provides a summary of several methods to create H2 from water using photocatalysis.  It also notes several strategies to store H2 in porous structures.  The current structure of this review article are problematic and after finishing each section of the review (H2 formation, H2 storage), I did not have better sense of the cutting edge strategies, successes, or future challenges with these two important catalytic areas. 

1) There are few figures and some are poor quality or not clear in what they are describing.  There are no equations, reactions, or tables in this review article.  I do not think the few number (6) of figures is sufficient to help the reader see the breadth of work in the photocatalysis field for H2 formation/storage.

2) This submission has well-written text, but it is all text.  In some cases it looks like each reference was given a two sentence summary and they were just combined to form a paragraph.  Many of the text sections felt disconnected.  More care needs to be taken to make the review text more than just an encyclopedic list of recent results.  A comparison table of key materials and outcomes are useful (material, light wavelength energy, quantitative H2 output, etc).  The "future challenges" section does not show that the authors have used their review and info to define clear areas for further study (versus restating the climate concerns noted in the introduction?). 

I hate to say it, but this submission feels like a general output from an AI program that had the references as the input that needed to be summarized.  There are many places where an undefined term from a paper is used, such as borospherene.  There could also be a much more quantitative results compared between different materials versus the heavily text based summary statements.

Comments on the Quality of English Language

There should be some additional attention paid to reorganizing paragraphs listing brief results on many different materials.  Are there better ways to organize the summary results?

Author Response

Revision details

Journal: Nanomaterials

Manuscript Title: Solar hydrogen production and storage in solid form: prospects for materials and methods

Manuscript ID: nanomaterials-3180978

Authors would like to thank the Editor and Reviewers for their insightful queries and suggestions. Based on the reviewer queries, the manuscript has been revised incorporating all the required contents. The author responses are highlighted in red color   both in author’s response and manuscripts

Reviewer 1

This review article provides a summary of several methods to create H2 from water using photocatalysis.  It also notes several strategies to store H2 in porous structures.  The current structure of this review article are problematic and after finishing each section of the review (H2 formation, H2 storage), I did not have better sense of the cutting edge strategies, successes, or future challenges with these two important catalytic areas. 

Reviewer query

1) There are few figures and some are poor quality or not clear in what they are describing.  There are no equations, reactions, or tables in this review article.  I do not think the few number (6) of figures is sufficient to help the reader see the breadth of work in the photocatalysis field for H2 formation/storage.

Author Response

We are very much thankful to the reviewer for his constrictive feedback of our review report. As suggested, the quality of the figures is improved, adding more figures. In this review we focused on a broad view focusing on two areas like (a) hydrogen generation methods and (b) storage methods concentrating solar hydrogen generation and solid-state storage. So, that new researchers, those who are interesting to involve on hydrogen energy can get some clear idea about hydrogen generation, transportation and storage. Moreover, if the topic is very confined focusing on materials and process, we can add more figures based on materials properties and processes. Since, this review is on general focus we could not add more figures.   However, we have added extra figures and the required contents including formulas.

Reviewer query

2) This submission has well-written text, but it is all text.  In some cases it looks like each reference was given a two sentence summary and they were just combined to form a paragraph.  Many of the text sections felt disconnected.  More care needs to be taken to make the review text more than just an encyclopedic list of recent results.  A comparison table of key materials and outcomes are useful (material, light wavelength energy, quantitative H2 output, etc).  The "future challenges" section does not show that the authors have used their review and info to define clear areas for further study (versus restating the climate concerns noted in the introduction?). 

I hate to say it, but this submission feels like a general output from an AI program that had the references as the input that needed to be summarized.  There are many places where an undefined term from a paper is used, such as borospherene.  There could also be a much more quantitative results compared between different materials versus the heavily text based summary statements.

There should be some additional attention paid to reorganizing paragraphs listing brief results on many different materials.  Are there better ways to organize the summary results?

Author Response

We understand your concern of the cogency and continuity of presentation. We revised the manuscript taking care of the presentation. As we stated earlier, since it has been focused on wide area we have consolidated the available results. If we try to review in detail about each and every work, it should be on specific topic, in this case it is difficult to confine with the certain pages of this wide area.

Comparison tables are included in the revised version of the manuscript for both solar hydrogen generation. As suggested the “future challenges” part has also been modified. The new terms like “borospherene” and others are taken care of and defined accordingly.  In case of AI, we don’t know how to use it for this purpose, and moreover we don’t like to use the AI for this purpose.

We have filly modified the manuscript as per your suggestions. Thanks for the valuable suggestions. 

Reviewer 2 Report

Comments and Suggestions for Authors

The draft by K. Adaikalam et al. is promising, as it refers to H2 production using solar power. In this regard, the manuscript can be a valuable addition to the body of reviews on this topic.

There are, however, a few aspects that need to be considered:

-          The abstract is too vague, as it contains almost no indication of cost, capacity, efficiency etc. for any of the materials / techniques mentioned;

-          The introduction is also too general, and there is considerable overlap between chapters 1 and 2; see for instance lines 70 and 113/114, and others.

-          Too much emphasis is placed on CO2 in the introduction, when the scope of this review is to focus on solar energy usage to obtain H2 from various sources, and to store (and transport) hydrogen to the end user in a meaningful and safe way;

-          Again, some data need to be included when talking about high energy density (how much?), abundance (..?) etc. Hydrogen also has more than 3 “colors”, and approaches (or even exceeds) 9 colors. As a review, the overall situation should be described properly.

-          There are several other reviews on solid-state storage of hydrogen, that can be cited in relation to this draft

-          Lines 128-129: petroleum vs gasoline use for gas reforming purposes should be mentioned;

-          Section (8) cannot stand on its own, being just a paragraph, that also just states some obvious phrases in lines 638-649, yet it contains no insight (what are the challenges? What are the migration strategies etc.).

-          Section (7) should be more detailed, and is probably could have been the focus of the whole draft. I recommend a graph summarizing the classes of H2 storage materials described herein. As the focus of the paper is on H2 storage, describing some theoretical models (DFT) such as that in ref. [91] and others, is less suitable. Ethylene is a gas, and that complex is neither cheap, nor easy to work with, and industry needs a cost-effective and safe material (authors also mention its use below 250K, so that already excludes its suitability);

-          Reports [99] –[101] could be described in more detail, given the practical applicability. The references [85], [105]-[106] etc, just like other DFT results, are meaningful only if applied under conditions set by DOE (i.e. not cryogenic etc)

-          Section 6.1. H2 can be stored by “(4) oxidation reactions with metal ion” . The authors need to give details about this process. This section is also very limited; several other mechanisms exist, some are presented in other sections (such as Section 7). I recommend reorganizing the information according to subchapters.

-          All Figures must be checked for copyright clearance, such as Fig 5 (is if production of the authors, was it taken from other publication(s) ?..)

-          Why is Figure 4 presented  under “Solar hydrogen storage”?

-          Statements like those from lines 475-478 are also too broad: “are being explored to identify efficient hydrogen storage solutions, yielding a range of results.” Are those results good, bad, what are the ranges obtained, citation of sources etc. etc.

-          Chapter 5 becomes too crowded with information, and a table needs to be drawn to summarize all the aspects described (type of material, storage capacity, experimental details aso.) Chemical formula needs to be included whenever possible , see for instrance line 303 :” introduced a novel complex perovskite”(which one?), line 306 (which halide perovskite ?) . There are numerous such omissions.

-          Chapters 5 and 7 should be subchapters of the same main section “Materials for hydrogen storage”. Considering the title, what is the justification for inclusion of chapter 7?

-          The draft needs to be proofread, there are some errors as well as font check (section 4.1)

-          The Conclusion section (9) is a reiteration of the ideas from the introduction/abstract; however, being a review of the state of the art regarding hydrogen production and storage, it should offer some insights regarding the prospects of the current technologies, what are the directions that need to be pursued, what is the current direction in the industrial landscape etc. etc. As it is, the conclusion is also quite vague and offers no real outlook;

-          As a general impression, there are too many general statements throughout this manuscript that may point out to a certain way in which the manuscript was produced, rather than devised. For instance, what are the actions governments are taking? Give at least one relevant example, picking at least one country, so that readers can understand where H2 storage is going at the moment.

Comments on the Quality of English Language

English language is mostly fine.

Author Response

Revision details

Journal: Nanomaterials

Manuscript Title: Solar hydrogen production and storage in solid form: prospects for materials and methods

Manuscript ID: nanomaterials-3180978

Authors would like to thank the Editor and Reviewers for their insightful queries and suggestions. Based on the reviewer queries, the manuscript has been revised incorporating all the required contents. The author responses are highlighted in red color   both in author’s response and manuscripts

Reviewer 2

The draft by K. Adaikalam et al. is promising, as it refers to H2 production using solar power. In this regard, the manuscript can be a valuable addition to the body of reviews on this topic.

There are, however, a few aspects that need to be considered:

Reviewer query

-          The abstract is too vague, as it contains almost no indication of cost, capacity, efficiency etc. for any of the materials / techniques mentioned;

Author Response

Here, in this review we have summarized the available works on both focusing solar hydrogen generation and solid state storage. As there is no successful methods and materials, so it is difficult to mention the cost, more than the cost, this new form of energy is need of the hour in this global worming condition. However, we have modified the abstract using relevant contents about cost and other queries.

Reviewer query

-          The introduction is also too general, and there is considerable overlap between chapters 1 and 2; see for instance lines 70 and 113/114, and others.

Author Response

Thanks for the suggestion, it has been modified suitably.

Reviewer query

-          Too much emphasis is placed on CO2 in the introduction, when the scope of this review is to focus on solar energy usage to obtain H2 from various sources, and to store (and transport) hydrogen to the end user in a meaningful and safe way;

Author Response

Here, in this work we have concentrated on solar hydrogen generation from water splitting and its storage in solid form. Since solar water splitting is free of CO2 emission and also globally all are trying to reduce CO2 emission, we thought to focus on this. And moreover, the solar water splitting is fully free from CO2 emission. For safe transport and utilization we are suggesting sloid storages of hydrogen and use by Solid oxide fuel cells.

Reviewer query

-          Again, some data need to be included when talking about high energy density (how much?), abundance (..?) etc. Hydrogen also has more than 3 “colors”, and approaches (or even exceeds) 9 colors. As a review, the overall situation should be described properly.

Author Response

Thanks again for the useful question, the energy density, storage capacity of hydrogen is mentioned in the revised manuscript. In case of abundance of hydrogen, it is understood from the nature, in every matter including water and hydrocarbons hydrogen is available, only thing is we must separate it as a useable form. In this respect, this review can give some ideas for the readers.

We have included a table   consolidating storage capacity of the works and materials available. 

 Reviewer query

-          There are several other reviews on solid-state storage of hydrogen, that can be cited in relation to this draft

Author Response

Most of the available reviews on hydrogen generation and storage are referred in this report other than solar hydrogen and solid storages.

Reviewer query

-          Lines 128-129: petroleum vs gasoline use for gas reforming purposes should be mentioned;

Author Response

It is also noted in the revised manuscript/

Reviewer query

-          Section (8) cannot stand on its own, being just a paragraph, that also just states some obvious phrases in lines 638-649, yet it contains no insight (what are the challenges? What are the migration strategies etc.).

Author Response

As suggested, the section (8) has been modified adding more points.

Reviewer query

-          Section (7) should be more detailed, and is probably could have been the focus of the whole draft. I recommend a graph summarizing the classes of H2 storage materials described herein. As the focus of the paper is on H2 storage, describing some theoretical models (DFT) such as that in ref. [91] and others, is less suitable. Ethylene is a gas, and that complex is neither cheap, nor easy to work with, and industry needs a cost-effective and safe material (authors also mention its use below 250K, so that already excludes its suitability);

Author Response

We agree with reviewer in his view, though, here we focused mainly on solar hydrogen and storages, we introduced the peripheral areas like hydrocarbons and their impact. The detailed report on other materials may increase the space and diluting the focused theme. However, the revised manuscript is suitable included these suggested materials.

Reviewer query

-          Reports [99] –[101] could be described in more detail, given the practical applicability. The references [85], [105]-[106] etc, just like other DFT results, are meaningful only if applied under conditions set by DOE (i.e. not cryogenic etc)

Author Response

As suggested, more points are added wherever possible from these reference in the revised manuscript.

Reviewer query

-          Section 6.1. H2 can be stored by “(4) oxidation reactions with metal ion” . The authors need to give details about this process. This section is also very limited; several other mechanisms exist, some are presented in other sections (such as Section 7). I recommend reorganizing the information according to subchapters.

Author Response

Thanks for the reviewer’s suggestion, accordingly we have modified the manuscript.

Reviewer query

-          All Figures must be checked for copyright clearance, such as Fig 5 (is if production of the authors, was it taken from other publication(s) ?..)

Author Response

Yes, checked, and for new figures are also we have obtained copyright permission.

Reviewer query

-          Why is Figure 4 presented  under “Solar hydrogen storage”?

Author Response

This figure is about different hydrogen storage methods, that is why we have included in solar hydrogen storage. Though, we have mainly concentrated on solid hydrogen storage, to differentiate and to mention advantages we have given symbolically all the storage techniques  

Reviewer query

-          Statements like those from lines 475-478 are also too broad: “are being explored to identify efficient hydrogen storage solutions, yielding a range of results.” Are those results good, bad, what are the ranges obtained, citation of sources etc. etc.

Author Response

The statements given in the lines 475-478 are “Effective hydrogen storage materials should possess good gravimetric and adsorption properties and low adsorption energy, allowing for easy desorption with minimal energy expenditure. Hydrides often require high temperatures for desorption, which can be problematic, while hydrates need high pressure to form”, it is just for introducing to the readers how are the hydrogen storage is affected. Yes, these are very broad topic, if we want to explain them detailed, it may be like to prepare a new review, if possible, we will do it in future.

Reviewer query

-          Chapter 5 becomes too crowded with information, and a table needs to be drawn to summarize all the aspects described (type of material, storage capacity, experimental details aso.) Chemical formula needs to be included whenever possible , see for instrance line 303 :” introduced a novel complex perovskite”(which one?), line 306 (which halide perovskite ?) . There are numerous such omissions.

Author Response

As suggested the manuscript has been revised adding table and chemical formulas and all

Reviewer query

-          Chapters 5 and 7 should be subchapters of the same main section “Materials for hydrogen storage”. Considering the title, what is the justification for inclusion of chapter 7?

Author Response

Our manuscript is are about solar hydrogen production and storage in solid form, we have changed this as “materials for solid hydrogen storage”

Reviewer query

-          The draft needs to be proofread, there are some errors as well as font check (section 4.1)

Author Response

Manuscript is thoroughly checked and corrected.

Reviewer query

-          The Conclusion section (9) is a reiteration of the ideas from the introduction/abstract; however, being a review of the state of the art regarding hydrogen production and storage, it should offer some insights regarding the prospects of the current technologies, what are the directions that need to be pursued, what is the current direction in the industrial landscape etc. etc. As it is, the conclusion is also quite vague and offers no real outlook;

Author Response

It is also modified as suggested

Reviewer query

-          As a general impression, there are too many general statements throughout this manuscript that may point out to a certain way in which the manuscript was produced, rather than devised. For instance, what are the actions governments are taking? Give at least one relevant example, picking at least one country, so that readers can understand where H2 storage is going at the moment.

Author Response

Based on reviewer suggestion, the relevant points are included in the revised manuscript.

Reviewer 3 Report

Comments and Suggestions for Authors

The presented review is focused on the solar hydrogen production and storage in solid form. In general, this work looks as a valuable study in this field. The structure of this review is logical and contains main topics related to the main subject. On the other hand there are several problems which should be solved before the acceptation.  The list of comments is provided below:
1.    It is difficult to find relevant elements in the text of the novelty of this review. There are other reviews in this field, e.g., Hassan et al. Energy Harvesting and Systems, 2024, 11, 20220134. Therefore authors should indicate how this review is deemed to be original with respect to the state-of-the-art in the field under consideration.
2.    Figure 3 looks too simple as the schematic presentation of photocatalytic water splitting process.
3.    Chapters 5 and 7 might be organized by introducing the table to better illustrate the content (discussion about the materials).

Comments on the Quality of English Language

The quality of English language is acceptable for the publication after minor editing.

Author Response

Revision details

Journal: Nanomaterials

Manuscript Title: Solar hydrogen production and storage in solid form: prospects for materials and methods

Manuscript ID: nanomaterials-3180978

Authors would like to thank the Editor and Reviewers for their insightful queries and suggestions. Based on the reviewer queries, the manuscript has been revised incorporating all the required contents. The author responses are highlighted in red color   both in author’s response and manuscripts

Reviewer 3

The presented review is focused on the solar hydrogen production and storage in solid form. In general, this work looks as a valuable study in this field. The structure of this review is logical and contains main topics related to the main subject. On the other hand there are several problems which should be solved before the acceptation.  The list of comments is provided below:

Reviewer query

  1. It is difficult to find relevant elements in the text of the novelty of this review. There are other reviews in this field, e.g., Hassan et al. Energy Harvesting and Systems, 2024, 11, 20220134. Therefore authors should indicate how this review is deemed to be original with respect to the state-of-the-art in the field under consideration.

Author Response

Thanks for the reviewer’s positive response on our articles. As suggested, the novelty and its originality of this work are presented in the modified form of the manuscript. And also, the suggested reference is referred in the modified manuscript. 

https://www.degruyter.com/document/doi/10.1515/ehs-2022-0134/html

Reviewer query

  1. Figure 3 looks too simple as the schematic presentation of photocatalytic water splitting process.

Author Response

This figure has been improved using modifications.

Reviewer query

  1. Chapters 5 and 7 might be organized by introducing the table to better illustrate the content (discussion about the materials).

Author Response

Thanks for the constructive suggestion, as suggested tables are included in the modified manuscript.

Round 2

Reviewer 1 Report

Comments and Suggestions for Authors

The authors have added several images and text revisions to clarify and improve their review.  Table 1 summarizing a range of storage systems is very useful for the reader and a similar table with selected summary of H2 generation structures and H2 output from different catalyst systems also seems necessary (and will break up the long stretches of dense text).  This could highlight the key results from some of the summarized systems described in the main text.

typo on line 333:  "excite" not "exit"

Author Response

Reviewer 1

Reviewer query

The authors have added several images and text revisions to clarify and improve their review.  Table 1 summarizing a range of storage systems is very useful for the reader and a similar table with selected summary of H2 generation structures and H2 output from different catalyst systems also seems necessary (and will break up the long stretches of dense text).  This could highlight the key results from some of the summarized systems described in the main text.

typo on line 333:  "excite" not "exit"

Author Response

We thank the reviewer very much for his positive remarks on our manuscript. And, we are sorry for blunder spelling mistake, and it is corrected in the revised form of manuscript.  The table suggested for solar hydrogen by water splitting is also included in the revised manuscript.

Reviewer 2 Report

Comments and Suggestions for Authors

Authors need to stay away from remarks like those at line 177-178: “It is almost equal to blue hydrogen.” A statement like that needs arguments, energy values, placing in real world context regarding energy production and consumption etc. None of these values are here presented to support this line. The colors of hydrogen should be placed in a diagram with colors and representation of color codes through the specific hydrogen type.

Figure 1 should be redrawn to better show the subsections of water splitting, thermochemical conversion and biological conversion.

AM1.5G from relation (1) is not defined anywhere in the draft.

In figure 3, correction is needed to properly show that H+ + 1e- -> ½ H2

Line 285: to efficiently function

“While this process is often considered expensive because of its energy requirements, it can become more cost-effective when powered by solar energy and proper recycling of used catalysts. “There is no reference or computation to support this statement. Also valid for lines 360-362.

Further on, photocatalysts could have been exemplified – there weren’t. And again, long phrases without references or concrete examples. This has to be fixed.

Last subsection of chapter 4 (4.3) is rather hard to read because there is no way to compare all those reports. I already suggested grouping these results in a table.

Figure 5 does not describe the types of solid storage hosts for hydrogen – it only shows pictures of such materials. Toluene and ammonia could also have their chemical structure represented, and the names could be better drawn so they can be read easier (i.e. horizontal writing).

Section 5 describes the ways in which H2 can be stored, not the mechanisms of H2 storage. For instance, please explain how someone reading section 5 could possibly understand how toluene is used for H2 storage? What is the mechanism?

Even section 5.1 has not even one chemical reaction showing production of H2!! Not to mention there is no regeneration of material presented. This is a serious omission.

“Hydrides often require high temperatures for desorption, which can be problematic, while hydrates need high pressure to form.” What hydrides? Which hydrates? No examples of any of these were given.

I still believe that DFT results cannot be included under “Materials for solid hydrogen storage”, because they have not been utilized as such. A separate subsection could be added for such theoretical investigation data.

“These hydrides can be ionic, covalent, volatile covalent, or metallic, depending on the nature of the metal and its interaction with hydrogen.” This cannot be one classification of hydrides, because more than one criterion is used. How is covalent related to a material being volatile? Which hydrides are volatile?

Table 1 already contains more DFT data than experimental. H2 needs to be produced experimentally, DFT simulations are a different story. There is no column for reversibility of these materials, which is essential for hydrogen storage.

Chapters 7 and 8 would need to be blended together. The conclusion section is too short as it is, and several references are lacking there (for the mentioned labs for instance). The overlap introduction – conclusion needs to be reduced.

Comments on the Quality of English Language

There are still many instances where misuse of English language was detected, even in the responses received from the authors. See for instance: "Yes, these are very broad topic, if we want to explain them detailed, it may be like to prepare a new review, if possible, we will do it in future."

Author Response

Reviewer 2

Authors need to stay away from remarks like those at line 177-178: “It is almost equal to blue hydrogen.” A statement like that needs arguments, energy values, placing in real world context regarding energy production and consumption etc. None of these values are here presented to support this line. The colors of hydrogen should be placed in a diagram with colors and representation of color codes through the specific hydrogen type.

Reviewer query

Figure 1 should be redrawn to better show the subsections of water splitting, thermochemical conversion and biological conversion.

Author Response

This figure is not included in the revision, it is included in the original form of manuscript. However, as asked, we have modified it using possible ways. Hydrogen generation can be given in wide areas based on fuels, energy sources and others. It is generated from a variety of materials from hydrocarbons, nonhydrocarbons and majorly from water-based biomasses using photo, thermal, electric, chemical, bio energies and their combinations. The major methods are given in modified figure 1.

Reviewer query

AM1.5G from relation (1) is not defined anywhere in the draft.

Author Response

For optical energy irradiation, a global standard known as AM1.5G is universally used. Solar simulators are classified in different spectral ranges AM1 (air mass 1), AM1.5 and AM2, among these largely used ranges is AM1.5 spectrum of atmospheric light intensity. Any source of light of its power equal to 1000 W/m2 or 100 mW/cm2 is known as AM1.5 G in global standards.

Reviewer query

In figure 3, correction is needed to properly show that H+ + 1e- -> ½ H2

Line 285: to efficiently function

Author Response

Yes, it is modified

Reviewer query

“While this process is often considered expensive because of its energy requirements, it can become more cost-effective when powered by solar energy and proper recycling of used catalysts. “There is no reference or computation to support this statement. Also valid for lines 360-362.

Author Response

As mentioned earlier, since there are no successful methods and materials, it is difficult to mention the cost, more than the cost, this new form of energy is need of the hour in this global worming conditions. However, we have modified the abstract using relevant contents about cost and others as stated here.

Though solar hydrogen generation seems to be costly, it is an energy-intensive process. Compared to other hydrogen generation methods, solar hydrogen generation by water splitting is green and comparatively cheap. If pure electrolysis of water is used, it is costly as it requires a source of energy, in solar water splitting solar energy is utilized. Only the initial cost may be high, once established, it will become sustained. Mainly, in the case of green hydrogen generation renewable energies such as wind and solar is used, which can reduce the cost.   According to Hydrogen Council, the cost of green hydrogen production ranged from $3.5 to $7.5 per kilogram. Though it seems to be costly, it depends on geographical location, transportation, storage and many factors. (https://fuelcellsworks.com/news/will-solar-hydrogen-fuel-cell-vehicles-become-the-norm)

Solar hydrogen and solid storage will be viable techniques. Considering the whole of solar hydrogen generation from water, solid storage and utilization is advanced considering other technics in all respects. Still, there are many efforts are going to reduce the cost and through technological advancements, its usage and increase application will make it a more competitive alternative to fossil fuels.

For the oxide materials mentioned (360-362), number of works are mentioned stating their efficiency in the manuscript in the text and tables.

Reviewer query

Further on, photocatalysts could have been exemplified – there weren’t. And again, long phrases without references or concrete examples. This has to be fixed.

Author Response

Thanks for the suggestion, we have modified the manuscript categorizing the materials. All statements given are referred. For example, in the sentences “Mehtab et al. reported a hydrogen generation rate of 467.2 μmol h1 g-cat1 for g-C3N4 nanosheets, which is higher than that of bulk g-C3N4, due to their efficient utilization of solar radiation. With an optical bandgap between 2.7 and 2.9 eV, g-C3N4 converts a significant portion of the solar spectrum into useful energy [58]”, these are only about same reference, and it is finally referred. Instead, every sentence need not be given reference number. 

Regarding classifications every paragraphs are separated to mention different class of materials for example the paragraph starts with “Metal oxide (MO) semiconductors…”, “Solid solutions…” and “Similarly, nitride materials..” etc. are to specify different kind of materials. I we want to give title for every group, it will be more subtitles. That’s why we have avoided further subheadings. 

Reviewer query

Last subsection of chapter 4 (4.3) is rather hard to read because there is no way to compare all those reports. I already suggested grouping these results in a table.

Author Response

Yes, as suggested, the possible reports are added to the table reducing the text sentences.

Reviewer query

Figure 5 does not describe the types of solid storage hosts for hydrogen – it only shows pictures of such materials. Toluene and ammonia could also have their chemical structure represented, and the names could be better drawn so they can be read easier (i.e. horizontal writing).

Author Response

It is to symbolically introduce the three types of storages, and as suggested the figure is modified in the revised version. The different methods of hydrogen storage and its related issues are mentioned in previous chapters and further it has been modified with references.

Reviewer query

Section 5 describes the ways in which H2 can be stored, not the mechanisms of H2 storage. For instance, please explain how someone reading section 5 could possibly understand how toluene is used for H2 storage? What is the mechanism?

Author Response

Here, in this review we have concentrated on solar hydrogen by water splitting and solid storage. The liquid form of storages is out of scope of this review, anyway the types are introduced to classify them. Hope the reviewer understands it.

Reviewer query

Even section 5.1 has not even one chemical reaction showing production of H2!! Not to mention there is no regeneration of material presented. This is a serious omission.

“Hydrides often require high temperatures for desorption, which can be problematic, while hydrates need high pressure to form.” What hydrides? Which hydrates? No examples of any of these were given.

Author Response

As we stated above and in an earlier response. It is consolidation of the works available on (1) hydrogen generation by solar water splitting and (b) solid storage of hydrogen. How all the materials and related kinetic reactions and all be described. If we have selected a single material or a group of materials for a single process, we can go for the detailed process and others.  It an overview of both the topics.

Reviewer query

I still believe that DFT results cannot be included under “Materials for solid hydrogen storage”, because they have not been utilized as such. A separate subsection could be added for such theoretical investigation data.

Author Response

As suggested the theoretical works are classified separately.

Reviewer query

“These hydrides can be ionic, covalent, volatile covalent, or metallic, depending on the nature of the metal and its interaction with hydrogen.” This cannot be one classification of hydrides, because more than one criterion is used. How is covalent related to a material being volatile? Which hydrides are volatile?

Author Response

These are introduced just to know the different properties of the materials. There is a difference in volatility of ionic and covalent compounds, the forces which keep covalent bonds are weaker than the forces which keep the lattice of ionic compounds together. Whereas the metal salts added organic solutions are volatile, which can be called volatile covalent compounds. For reviewer’s information, there is a separate work on volatile covalent hydrides “Study of the conditions for generating volatile covalent hydrides of tin from solutions of Sn2+ and Sn4+ and atomization in a silica tube” (Microchemical Journal, 39, 119-125 (1989)). That is covalent forces are also weakened due to surficial interactions.

Reviewer query

Table 1 already contains more DFT data than experimental. H2 needs to be produced experimentally, DFT simulations are a different story. There is no column for reversibility of these materials, which is essential for hydrogen storage.

Author Response

Yes, as stated above the theoretical and experimental results are separated.

Reviewer query

Chapters 7 and 8 would need to be blended together. The conclusion section is too short as it is, and several references are lacking there (for the mentioned labs for instance). The overlap introduction – conclusion needs to be reduced.

Author Response

Thanks for the suggestion, as suggested they are blended.

Round 3

Reviewer 2 Report

Comments and Suggestions for Authors

In Figure 3, all reactions depicted should be balanced, i.e. coefficients need to be added. This has not been thoroughly fixed. Same issue in line 286. The problem persists in Fig. 4.

Reaction on line 286 is also missing reaction participants. All reactions need to be numbered: (1), (2) etc.

In some paragraphs the words are used relentlessly; see lines 345-347,: “efficient”  is present 3 times in that sentence. Reduce the number of uses to strictly necessary.

Some repetitions need to be removed – see the info on the work of Fujishima and Honda that is mentioned twice under same section, in relation to TiO2 electrode used.

Figure 5 still doesn’t present the proper names for all involved solid-state hydrogen storage materials depicted. This needs to be fixed (see above, ammonia or toluene are given with names, toluene has even its molecular formula shown).

Since data on absorption energy is scarce (only 1 entry), Table 2 could contain this column merged with H2 release temperature. There is a distinction between N/A and no absorption energy; the authors should carefully decide which is the case for the 6 entries in Table 6.

Lines 833-834: Some phrases need more polish to have sense: “If pure electrolysis of water is used, it is costly as it requires a source of energy, in solar water splitting solar energy is utilized.”

Overall, the language needs serious brushing, perhaps from a native speaker; see another such example: “Considering the combined strategy of solar hydrogen generation from water, solid storage and utilization, it is advanced considering other technics in all respects..” on lines 846-847.

Global agencies mentioned under conclusion should be referred to using some literature citation or site link under References section.

Comments on the Quality of English Language

ines 833-834: Some phrases need more polish to have sense: “If pure electrolysis of water is used, it is costly as it requires a source of energy, in solar water splitting solar energy is utilized.”

Overall, the language needs serious brushing, perhaps from a native speaker; see another such example: “Considering the combined strategy of solar hydrogen generation from water, solid storage and utilization, it is advanced considering other technics in all respects..” on lines 846-847.

Author Response

Revision details

Journal: Nanomaterials

Manuscript Title: Solar hydrogen production and storage in solid form: prospects for materials and methods

Manuscript ID: nanomaterials-3180978

We thank the Editor and Reviewers for their useful suggestion to improve the quality of our manuscript. Based on the reviewer queries, the manuscript has been revised incorporating all the required contents considering maximum care. The author responses are highlighted in red color   both in author’s response and manuscripts

Reviewer query

In Figure 3, all reactions depicted should be balanced, i.e. coefficients need to be added. This has not been thoroughly fixed. Same issue in line 286. The problem persists in Fig. 4.

Reaction on line 286 is also missing reaction participants. All reactions need to be numbered: (1), (2) etc.

Author Response

Thanks for the suggestion, the stated equation is corrected and accordingly the manuscript has also been modified. The Figure 4 also corrected mentioning all the figures in the text.

Reviewer query

In some paragraphs the words are used relentlessly; see lines 345-347,: “efficient”  is present 3 times in that sentence. Reduce the number of uses to strictly necessary.

Some repetitions need to be removed – see the info on the work of Fujishima and Honda that is mentioned twice under same section, in relation to TiO2 electrode used.

Author Response

Thanks for the formation, these mistakes are all corrected with native speaker’s help.

Reviewer query

Figure 5 still doesn’t present the proper names for all involved solid-state hydrogen storage materials depicted. This needs to be fixed (see above, ammonia or toluene are given with names, toluene has even its molecular formula shown).

Author Response

It is given to symbolically mention and differentiate the solid storage principle that is physisorption and chemisorption from other methods. It has been revised mentioning the types of adsorptions labelling the figures as other methods.

Reviewer query

Since data on absorption energy is scarce (only 1 entry), Table 2 could contain this column merged with H2 release temperature. There is a distinction between N/A and no absorption energy; the authors should carefully decide which is the case for the 6 entries in Table 6.

Author Response

It has been corrected as suggested.

Reviewer query

Lines 833-834: Some phrases need more polish to have sense: “If pure electrolysis of water is used, it is costly as it requires a source of energy, in solar water splitting solar energy is utilized.”

Author Response

It is carefully checked and modified.

Reviewer query

Overall, the language needs serious brushing, perhaps from a native speaker; see another such example: “Considering the combined strategy of solar hydrogen generation from water, solid storage and utilization, it is advanced considering other technics in all respects..” on lines 846-847.

Author Response

The whole manuscript has been checked by native speaker thoroughly.

Reviewer query

Global agencies mentioned under conclusion should be referred to using some literature citation or site link under References section.

Author Response

Yes, as suggested possible references are included with expansion of abbreviated names.  
